# Epidemiology and Clinical Characteristics Based on the Rome III and IV Criteria of Japanese Patients with Functional Dyspepsia

**DOI:** 10.3390/jcm11092342

**Published:** 2022-04-22

**Authors:** Sota Aono, Toshihiko Tomita, Katsuyuki Tozawa, Daisuke Morishita, Keisuke Nakai, Takuya Okugawa, Masashi Fukushima, Tadayuki Oshima, Hirokazu Fukui, Hiroto Miwa

**Affiliations:** 1Division of Gastroenterology and Hepatology, Department of Internal Medicine, Hyogo College of Medicine, Nishinomiya 663-8501, Japan; so-aono@hyo-med.ac.jp (S.A.); tomita@hyo-med.ac.jp (T.T.); da-morishita@hyo-med.ac.jp (D.M.); k-nakai@hyo-med.ac.jp (K.N.); okugawat@hyo-med.ac.jp (T.O.); ma-fukushima@hyo-med.ac.jp (M.F.); t-oshima@hyo-med.ac.jp (T.O.); hfukui@hyo-med.ac.jp (H.F.); 2Department of Gastroenterology and Hepatology, Amagasaki Chuo Hospital, Amagasaki 661-0976, Japan; katu-you@chuoukai.or.jp

**Keywords:** functional dyspepsia, Rome criteria, epigastric symptoms

## Abstract

The subtypes of functional dyspepsia (FD) differ depending on whether the Rome III criteria or the Rome IV criteria are used. We investigated the ability to diagnose FD patients using the Rome III and IV criteria. The subtypes of FD were evaluated using the Rome questionnaire. The Gastrointestinal Symptom Rating Score, health-related quality of life (HR-QOL; SF-8), and psychological scores (HADS, STAI) were evaluated. The questionnaire was collected from a total of 205 patients, and 54.1% were FD patients. The ratio of FD patients under the Rome III criteria was 19% for epigastric pain syndrome (EPS), 38% for postprandial distress syndrome (PDS), and 43% for an overlap of EPS and PDS, but under the Rome IV criteria overlap decreased to 17% and PDS increased to 64%. Patients whose subtype changed from overlap under the Rome III criteria to PDS under the Rome IV criteria were compared with PDS patients whose subtype did not change between the Rome III and IV criteria. The comparison showed that the former had significantly lower early satiation rates and significantly higher acid reflux and abdominal pain scores, demonstrating that EPS symptoms due to acid reflux after meals were clearly present. As a result of changing from the Rome III criteria to the Rome IV criteria, the number of overlap patients decreased, and the number of PDS patients increased.

## 1. Introduction

Functional dyspepsia (FD) is a disease in which epigastric symptoms, such as upset stomach and stomach pain, continue chronically even though no organic disease is present [1,2]. The Rome diagnostic criteria are used internationally to classify FD [3,4,5,6,7]. Under the Rome III criteria, dyspepsia symptoms are narrowed down to four symptoms (postprandial fullness, early satiation, epigastric pain, and epigastric burning), and the duration of the symptoms is defined as “having had symptoms for 6 months or longer and have continuing symptoms for the past 3 months.” Of these, cases of meal-related symptoms, such as postprandial fullness and early satiation, are defined as postprandial distress syndrome (PDS), and cases of meal-unrelated symptoms, such as epigastric pain and burning, are classified as epigastric pain syndrome (EPS), but it has been reported that the symptoms often overlap [8,9,10,11].

The “actual Rome IV criteria” added several items to support the Rome III criteria. We also made some revisions, such as adding the term “bothersome” to the four typical symptoms to indicate severity [7]. However, generally speaking, it is the same as the Rome III criteria; thus, the subtype was not affected by the change to the actual Rome IV criteria. In this study, we focused on epigastric symptoms induced after eating and adopted the criteria proposed by Tack et al. as the Rome IV criteria [10,12,13]. Under the Rome IV criteria, patients with postprandial symptoms were all classified into the PDS group.

For example, a patient who complains of epigastric pain after eating is classified as overlap PDS-EPS under the Rome III criteria, but as PDS under the Rome IV criteria; thus, situations occur in clinical practice where the type of FD differs depending on which criteria are used. A recent study from Belgium reported that when the same patients were classified by both the Rome III and IV criteria, use of the Rome IV criteria led to a decrease in the overlap group and an increase in the PDS group compared to the Rome III criteria [12]. However, to date, there have been no reports examining the prevalence and classification rate of FD patients using the Rome IV criteria in Japan, and the clinical significance of the appropriate use of the Rome III criteria and IV criteria remains unknown.

Therefore, the aims of the present study were to perform FD diagnosis using both the Rome III and IV criteria and investigate the diagnostic ability and the characteristics of background factors under both sets of criteria.

## 2. Materials and Methods

### 2.1. Study Design

The present study was a multicenter, prospective, observational study conducted at our hospital and 8 related facilities. A questionnaire survey on gastrointestinal symptoms, health-related quality of life (HR-QOL), and psychological factors was conducted among patients who complained of epigastric symptoms, visited an outpatient clinic, and in whom organic disease could be ruled out by esophagogastroduodenoscopy (EGD), etc. from April 2020 to October 2020. All subjects received sufficient explanation orally concerning the details of the study at the time of consultation, and written, informed consent to participate in the present study was obtained from all subjects. The present clinical trial was approved by the institutional review board of Hyogo College of Medicine (approval number 3467). Patients who gave their informed consent were enrolled in this study. The UMIN registration number is UMIN000041094. The present study was conducted according to the principles governing human research in the Declaration of Helsinki. All authors had access to the study data and reviewed and approved the final manuscript.

### 2.2. Patients

All patients met the following criteria: outpatients aged 20 to 75 years, EGD performed within 1 year of consultation to exclude organic disease as a cause of epigastric symptoms. Findings such as chronic gastritis and duodenitis were classified as nonorganic upper digestive disease. The present study excluded patients with a history of upper gastrointestinal surgery (endoscopic treatment was permitted), and patients with clear causes of upper digestive symptoms such as malignancy and digestive ulcers, neurological diseases such as Parkinson’s disease, and metabolic diseases such as diabetes mellitus. Furthermore, patients taking drugs such as steroids, GI motility drugs, antiacids, prostaglandine preparations, antidepressants, anxiolytics, and antipsychotics were also excluded. In addition, patients were excluded if they had tested positive for *Helicobacter pylori*, if they had undergone *H. pylori* eradication therapy within 6 months of initial consultation, or if they were currently undergoing eradication therapy. *H. pylori* infection was diagnosed by evaluating the serum HP-IgG of patients with atrophic gastritis observed via EGD. Patients with significant IBS were also excluded, including those whose epigastric pain was relieved by defecation or passing gas [14,15]. Additionally, patients with significant GERD were excluded [16,17].

### 2.3. Assessment

The Japanese version of Rome III (partially modified) was used for FD diagnosis. FD was diagnosed in patients who had no obvious organic disease that caused the symptoms, had experienced the symptoms for at least 6 months, and had the symptoms for the previous 3 months (symptom persistence was defined as occurring 2–3 times or more per week in the case of PDS and at least once per week in the case of EPS). PDS under the Rome III criteria is indicated by meal-related symptoms such as postprandial fullness and early satiation, and EPS under the Rome III criteria is indicated by meal-unrelated symptoms such as epigastric pain and burning; the overlap in the Rome III criteria was the group of patients who had both PDS and EPS symptoms under the Rome III criteria. In this study, the criteria proposed by Tack et al. were adopted as the Rome IV criteria; under these criteria, all postprandial epigastric symptoms were classified as PDS, according to the previous study [10,12,13]. Specifically, under the Rome IV criteria, all patients with very postprandial epigastric symptoms (classified as the overlap under the Rome III criteria) were classified as PDS. Assessment items that were examined to determine whether they had an effect were: age, sex, body mass index (BMI), smoking history, drinking history, H. pylori eradication history, epigastric symptoms (i.e., postprandial fullness, early satiation, epigastric symptoms, postprandial epigastric symptoms, heartburn), gastrointestinal symptoms based on a self-administered questionnaire, HR-QOL, and anxiety and depression scores. In addition, all questions related to symptoms were asked under the assumption that the symptoms were bothersome.

### 2.4. Questionnaires (Symptoms, QOL, Anxiety, and Depression)

Gastrointestinal symptoms were assessed using the Japanese version of the gastrointestinal symptom rating scale (GSRS) questionnaire. The GSRS consists of 15 questions, each of which falls under one of five subscales (i.e., reflux, abdominal pain, dyspepsia, diarrhea, constipation). The responses for each item ranged from a score of 1 to 7, where 1 indicated that the gastrointestinal symptoms had no effect on the patient’s activities of daily living, and 7 indicated that the gastrointestinal symptoms have an unacceptable adverse effect on the patient’s activities of daily living. It is an index that uses the mean value of each item as the subscale score and the mean value of all subscales as the total score [18]. The SF-8 was used to assess HR-QOL. In the SF-8, the following eight health concepts are measured: physical functioning (PF), role physical (RP), bodily pain (BP), general health (GH), vitality (VT), social functioning (SF), role emotional (RE), and mental health (MH) [19]. The hospital anxiety and depression scale (HADS) was used to assess anxiety and depression. The HADS consists of 14 items (7 items concerning anxiety and 7 items concerning depression) and is used to evaluate anxiety, depression, and total scores. Higher scores were considered to represent a higher degree of psychological distress [20]. In addition, the state-trait anxiety inventory (STAI) was used to assess anxiety, including state anxiety, which is anxiety dependent on the situation, and trait anxiety, which indicates having a personality that is usually evasive of or tends to worry about risk, with higher scores indicating more severe degrees of anxiety [21]. In the present study, all items were evaluated once at the time of consultation, and their effects were investigated.

### 2.5. Statistical Analysis

All results are expressed as means ± standard deviation (SD). The paired t-test, Mann–Whitney U test, and Fisher’s exact test were used for comparison of the two groups, as appropriate. Statistical significance was defined as a value of *p* < 0.05. Statistical analysis was performed using GraphPad Prism 5 (GraphPad Software, La Jolla, CA, USA).

## 3. Results

### 3.1. Enrolment and Baseline Characteristics of the Patients

There were 210 patients who complained of epigastric symptoms, visited the outpatient department, and from whom a questionnaire could be collected. Of these, 5 were positive for H. pylori, 0 had undergone H. pylori eradication therapy within 6 months of initial consultation, and 0 were currently undergoing eradication therapy. There were no subjects who did not respond to the questionnaire, and a total of 205 patients, excluding 5 H. pylori-positive patients, were included in the final analysis. The target FD patients in the present study were 111 patients in whom FD could be diagnosed based on the Rome III and IV criteria (Figure 1). The prevalence of FD was 54.1% (111/205). The mean age of FD patients was 54.3 ± 14.9 years, and the male/female ratio was 35/76 (68.5% female). Patients’ characteristics are shown in Table 1.

### 3.2. FD Subgroups under the Rome III Criteria vs. the Rome IV Criteria

Under the Rome III criteria 21 patients (19%) had EPS, 42 (38%) had PDS, and there was an overlap of PDS-EPS in 48 (43%). Under the Rome IV criteria, there was no difference in the frequency of EPS, but PDS increased to 71 persons (64%) and the overlap of PDS-EPS decreased to 19 (17%) (Figure 2).

### 3.3. Patient Background Factors for FD under the Rome III Criteria vs. the Rome IV Criteria

There were no significant differences between the PDS group diagnosed by the Rome III criteria and the PDS group diagnosed by the Rome IV criteria in patient background factors, epigastric symptoms, HR-QOL, psychological score, and gastrointestinal symptoms. In addition, no significant differences were observed between the overlap group diagnosed by the Rome III criteria and the overlap group diagnosed by the Rome IV criteria.

### 3.4. Characteristics of PDS to PDS vs. Overlap to PDS

Of the 71 FD patients (64%) who received a diagnosis of PDS under the Rome IV criteria, 42 (38%) received a diagnosis of PDS based on both the Rome III criteria and IV criteria (PDS to PDS). Moreover, the classification of 29 patients (26%) changed from PDS and EPS under the Rome III criteria to PDS under the Rome IV criteria (overlap to PDS) (Figure 3).

No significant differences were observed between the two groups in the background factors of age, sex, BMI, history of smoking, history of alcohol intake, and rate after H. pylori eradication. There were no significant differences in postprandial fullness, chest pain, or heartburn as epigastric symptoms, but early satiation appeared significantly more frequently in the group of patients diagnosed with PDS under both the Rome III and IV criteria (73.8% and 41.4%, *p* < 0.01) (Table 2).

### 3.5. QOL, Psychological Score, Gastrointestinal Symptoms (PDS to PDS vs. Overlap to PDS)

Evaluations using the SF-8, which is a general QOL scale, demonstrated that the two groups did not show any significant differences in PF, RP, GH, VT, SF, RE, MH, PCS, and MCS, but QOL regarding BP was significantly lower in the group of patients whose classification changed from overlap under the Rome III criteria to PDS under the Rome IV criteria (46.3 ± 8.8 and 41.5 ± 9.5, *p* < 0.05) (Table 3). Regarding the HADS, no significant differences were observed between the two groups in the anxiety score, depression score, or total score.

Concerning the STAI, there were no significant differences in state anxiety or trait anxiety. In addition, no significant differences were observed between the two groups in the GSRS questionnaire, dyspepsia score, diarrhea score, constipation score, or total score, but the acid reflux score was significantly higher in the group of patients whose classification changed from overlap under the Rome III criteria to PDS under the Rome IV criteria (2.8 ± 1.4 and 3.6 ± 1.6, *p* < 0.05). The abdominal pain score was significantly higher in the group of patients whose classification changed from overlap under the Rome III criteria to PDS under the Rome IV criteria (2.9 ± 1.2 and 3.7 ± 1.4, *p* < 0.05) (Table 4).

## 4. Discussion

In the present study, a self-administered questionnaire survey including gastrointestinal symptoms, HR-QOL, and psychological factors was conducted among FD patients who visited our hospital and related facilities for dyspepsia symptoms. This was a multicenter, prospective, observational study examining the diagnostic ability and differences of subtypes of FD patients in the Rome III and IV criteria. To date, there have been no reports involving Asian FD patients that have examined the subtypes and characteristics of FD using both the Rome III and IV criteria at the same time in the same patients. The results of the present study showed that the overlap group decreased and the PDS group increased under the Rome IV criteria when compared to the Rome III criteria.

The prevalence of FD in the present study was 54.1%, and it is generally reported that the prevalence of FD in patients who visited a hospital complaining of dyspepsia is approximately half of cases [21]. Thus, the results of the present study were consistent with previous studies. Under the Rome III criteria, meal-related epigastric symptoms such as postprandial fullness and early satiation, and meal-unrelated epigastric symptoms such as epigastric pain and burning, are classified as separate subtypes [1,2,5]. However, since the epigastric symptoms often overlap in FD patients, it is difficult to determine the subtype of FD patients, and it is often difficult to select treatment.

Using the criteria proposed by Tack et al. as the Rome IV criteria, when analyzing the meal-related symptoms, rather than the types of symptoms themselves, we have defined that all patients who complain of epigastric symptoms after eating are to be classified as having PDS [10,12,13]. A recent study by Carborn et al. in Belgium examined FD subtypes in the same patients using the Rome III and IV criteria, and they found that overlap PDS-EPS patients decreased from 61% to 18%, and PDS patients increased from 30% to 73% [12]. There are no previous prospective, observational studies using the Rome III and IV criteria for Japanese FD patients, and the present study is the first study comparing the Rome III and IV criteria in Japanese FD patients. It was found that PDS and EPS overlapped in approximately half (48/111) of patients based on the Rome III criteria, and the change from the Rome III criteria to the Rome IV criteria led to a reduction in overlap PDS-EPS patients from 43% to 17% and an increase in PDS patients from 38% to 64%. Specifically, this shows that, under the Rome IV criteria, it has become possible to classify PDS and EPS subtypes more clearly by stipulating meal-related symptoms.

If that is the case, in what type of patients did the subtype of FD change from the Rome III criteria to the Rome IV criteria? First, the clinical background factors of PDS patients based on the Rome III criteria and those of PDS patients based on the Rome IV criteria were compared, but no significant difference was noted regarding any of the items. These results suggest that the reason why significant differences were not noted for each patient background item might have been because approximately 60% of the PDS patients diagnosed based on the Rome IV criteria in the present study were PDS patients diagnosed based on the Rome III criteria for comparison. 

Next, we compared the clinical characteristics of patients whose subtypes did not change and those who did change under the Rome III and IV criteria. The results of the comparison of 42 patients in the PDS group (PDS to PDS) based on both the Rome III and IV criteria, and 29 patients whose subtype changed from overlap PDS-EPS under the Rome III criteria to PDS under the Rome IV criteria (overlap to PDS), showed that in the group whose subtype changed from overlap to PDS, early satiation and QOL related to physical pain were significantly lower, and regarding gastrointestinal symptoms, the acid reflux score and abdominal pain score were significantly higher. These findings suggest that, in the group in which the subtype changed from overlap under the Rome III criteria to PDS under the Rome IV criteria, EPS symptoms due to postprandial acid reflux might have been more prominent than PDS symptoms such as early satiation. In pharmacotherapy for FD patients, it is important to classify the cases as PDS, EPS, or overlap, and it is generally recommended in the guidelines to treat PDS patients with prokinetics and EPS patients with inhibitors of gastric acid secretion [22,23,24,25,26,27,28,29]. Since the results of the present study suggested that, in the group in which the subtype changed from overlap under the Rome III criteria to PDS under the Rome IV criteria, there might have been severe postprandial EPS symptoms, especially acid reflux symptoms, it may be more appropriate to use inhibitors of gastric acid secretion rather than prokinetics in such a patient group. The results of the present study demonstrated that diagnosing subtypes in patients who complained of dyspepsia by using two classifications (the Rome III and IV criteria) at the same time, FD subtypes that could previously only be classified into three subtypes (i.e., PDS, EPS, overlap) could be classified into four subtypes (i.e., PDS to PDS, EPS, overlap to overlap, overlap to PDS). This may be of assistance in the selection of therapeutic agents in clinical practice. A recent study by Van den Houte et al. reported that patients who received a diagnosis of overlap based on both the Rome III and IV criteria (overlap to overlap) consisted of a group of patients closer to PDS than EPS; therefore, the prescription of prokinetics was recommended [30]. On the basis of the results of the present study, prokinetics should be used as first-line therapy in groups that receive a diagnosis of PDS or overlap based on both the Rome III and IV criteria, and inhibitors of gastric acid secretion should be used as first-line therapy for patients whose subtype changes from overlap under the Rome III criteria to PDS under the Rome IV criteria. However, at present, there are no reports of the therapeutic effects for each subtype under the Rome IV criteria; therefore, it is necessary to examine the therapeutic effects of drugs for patients in each subtype in the future and accumulate further evidence.

The present study had several limitations. First, compared with other epidemiological studies of FD, the sample size was relatively small and data were collected only from Japanese patients. Second, whereas the present study was a multicenter, prospective observational study, of the 205 patients who responded to the questionnaire, 140 (68.3%) were from a university hospital, and as such there might have been bias between facilities. Due to the fact that numerous patients who were visiting a university hospital also had a long disease duration, the results of the study might have been affected. In addition, the age of FD patients enrolled in the present study was approximately 55 years, which is older than that of generally reported FD patients, and this may be related to there being numerous patients enrolled in the study being treated in university hospitals. Furthermore, in this study, the questionnaire survey was limited to patients with epigastric symptoms; thus, the symptoms of PDS patients were restricted to the upper abdomen. However, the diagnosis of PDS according to the Rome criteria does not necessarily require the presence of symptoms in the upper abdomen, which may have resulted in a lower frequency of PDS than if the actual Rome criteria had been applied. Finally, this study obtained answers from participants using a questionnaire; therefore, we cannot dismiss the possibility that the responses were influenced by recall bias. Nonetheless, the patient-reported outcomes were analyzed; as such, we consider the results to be reliable.

## 5. Conclusions

The present study is the first multicenter prospective observational study in Japan to examine the subtype and clinical characteristics of FD patients using the Rome III and IV criteria in patients who visited a hospital complaining of epigastric symptoms. As a result of changing from the Rome III criteria to the Rome IV criteria, the number of overlap patients decreased, the number of PDS patients increased, and it was possible to classify cases of PDS and EPS more clearly. Our findings also suggested that using the Rome III and IV criteria in FD diagnosis at the same time may be of assistance in the selection of therapeutic agents in clinical practice.

## Figures and Tables

**Figure 1 jcm-11-02342-f001:**
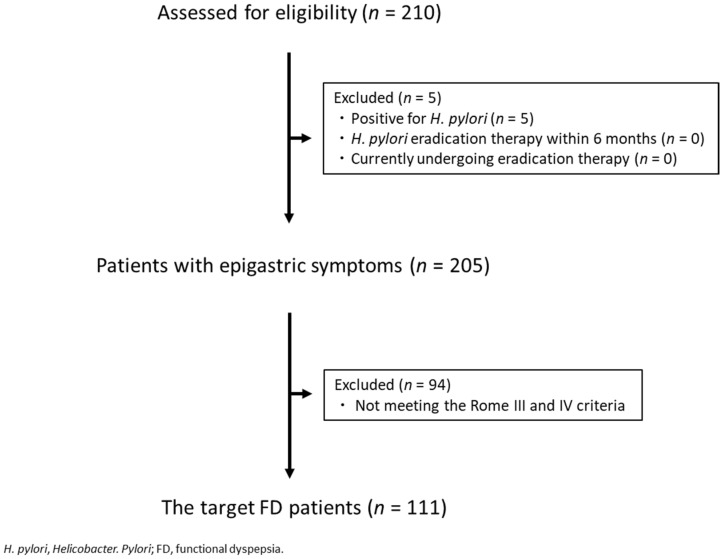
CONSORT chart.

**Figure 2 jcm-11-02342-f002:**
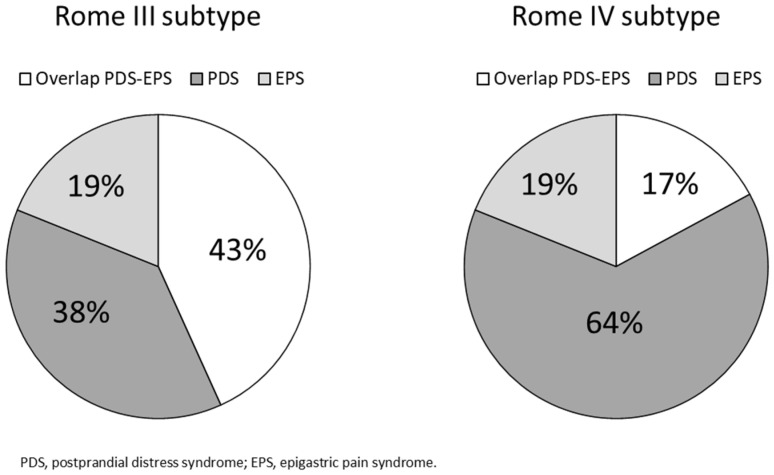
Proportions of functional dyspepsia subtypes by the Rome III and IV criteria.

**Figure 3 jcm-11-02342-f003:**
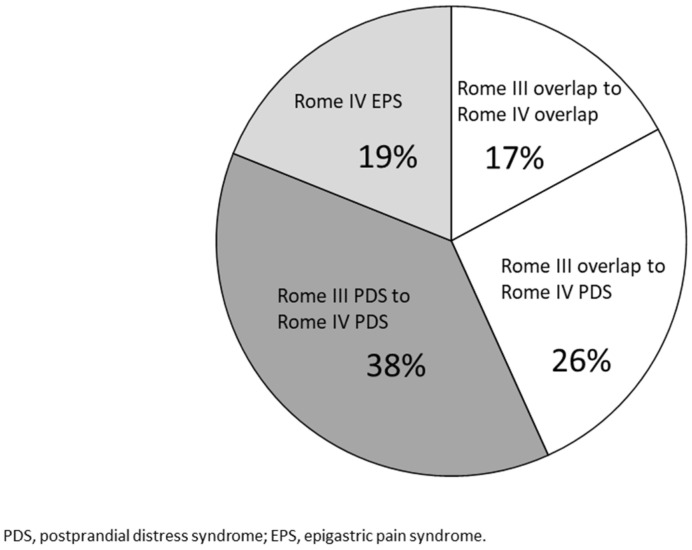
Combination of the Rome III and IV criteria. Overlap to PDS: patients whose diagnosis was changed from overlap of EPS and PDS in the Rome III criteria to PDS in the Rome IV criteria. PDS to PDS: patients diagnosed with PDS in both the Rome III and IV criteria. Overlap to overlap: Patients diagnosed with overlap in both the Rome III and IV criteria.

**Table 1 jcm-11-02342-t001:** Characteristics and symptoms of the patients.

	FD	Rome III			Rome IV		
		Overlap PDS-EPS	PDS	EPS	Overlap PDS-EPS	PDS	EPS
*n*	111	48	42	21	19	71	21
Age (y)	54.3 ± 14.9	51.8 ± 15.5	57.3 ± 12.4	53.8 ± 16.9	55.5 ± 13.1	54.1 ± 14.7	53.8 ± 16.9
Gender (% female)	68.5	70.8	69	61.9	52.6	74.6	61.9
BMI (kg/m^2^)	21.8 ± 3.5	22.2 ± 3.6	21.4 ± 3.7	21.9 ± 2.6	22.7 ± 4.1	21.6 ± 3.5	21.9 ± 2.6
Smoking	23.4%	25%	16.7%	33.3%	31.6%	18.3%	33.3%
Drinking	28.8%	27.1%	19%	52.4%	26.3%	22.5%	52.4%
After eradication of *Helicobacter pylori*	22.5%	22.9%	14.3%	33.3%	26.3%	18.3%	33.3%
Postprandial fullness	54.1%	72.9%	59.5%	0%	84.2%	62%	0%
Early satiation	46.8%	43.8%	73.8%	0%	47.4%	60.6%	0%
Epigastric pain or burning	61.3%	100%	0%	100%	100%	40.8%	100%
Postprandial epigastric pain or burning	27%	60.4%	0%	0%	0%	40.8%	0%
Chest pain	60.4%	64.6%	40.5%	66.7%	74%	47.9%	66.7%
Heartburn	51.4%	64.6%	40.5%	42.9%	74%	47.9%	42.9%

FD, functional dyspepsia; PDS, postprandial distress syndrome; EPS, epigastric pain syndrome; BMI, body mass index.

**Table 2 jcm-11-02342-t002:** Characteristics and symptoms between the Rome III and IV criteria.

Characteristics and Symptoms	Rome III to Rome IV		*p* Value
	PDS to PDS	Overlap to PDS	
Patients (*n*)	42	29	
Age (y)	57.3 ± 12.4	49.3 ± 16.5	0.050
Gender (% female)	69	82.8	0.198
BMI (kg/m²)	21.4 ± 3.7	21.8 ± 3.2	0.717
Smoking	16.7%	20.7%	0.675
Drinking	19%	27.6%	0.405
After eradication of *Helicobacter pylori*	14.3%	24.1%	0.299
Postprandial fullness	59.5%	65.5%	0.617
Early satiation	73.8%	41.4%	0.007
Epigastric pain or burning	0%	100%	
Postprandial epigastric pain or burning	0%	100%	
Chest pain	40.5%	62.1%	0.236
Heartburn	40.5%	58.6%	0.137

PDS, postprandial distress syndrome; BMI, body mass index.

**Table 3 jcm-11-02342-t003:** HR-QOL (PDS to PDS vs. overlap to PDS).

SF-8	PDS to PDS	Overlap to PDS	*p* Value
Patients (*n*)	42	29	
PF	45.6 ± 9.4	42.8 ± 10.1	0.157
RP	44.0 ± 9.5	42.6 ± 10.2	0.484
BP	46.3 ± 8.8	41.5 ± 9.3	0.025
GH	40.2 ± 7.1	39.1 ± 8.2	0.358
VT	42.1 ± 9.0	43.1 ± 7.4	0.712
SF	41.4 ± 11.2	39.2 ± 12.1	0.469
RE	44.1 ± 9.2	42.3 ± 11.7	0.641
MH	42.7 ± 7.2	42.8 ± 8.2	0.808
PCS	43.9 ± 8.0	40.8 ± 9.7	0.124
MCS	41.6 ± 9.0	42.0 ± 8.5	0.949

PDS, postprandial distress syndrome; EPS, epigastric pain syndrome; SF-8, 8-item short-form health survey; PF, physical functioning; RP, role physical; BP, bodily pain; GH, general health perception; VT, vitality; SF, social functioning; RE, role emotional; MH, mental health; PCS, physical component summary; MCS, mental component summary.

**Table 4 jcm-11-02342-t004:** GSRS (PDS to PDS vs. overlap to PDS).

GSRS	PDS to PDS	Overlap to PDS	*p* Value
Patients (*n*)	42	29	
Reflux	2.8 ± 1.4	3.6 ± 1.6	0.025
Abdominal pain	2.9 ± 1.2	3.7 ± 1.4	0.024
Dyspepsia	2.9 ± 1.2	2.8 ± 1.3	0.449
Diarrhea	2.2 ± 1.3	2.8 ± 1.8	0.138
Constipation	3.1 ± 1.4	2.7 ± 1.2	0.245
Total score	2.8 ± 0.90	3.1 ± 1.1	0.259

PDS, postprandial distress syndrome; EPS, epigastric pain syndrome; GSRS, gastrointestinal symptom rating scale

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
