# Peer review of "Epidemiology and Clinical Characteristics Based on the Rome III and IV Criteria of Japanese Patients with Functional Dyspepsia"

_jcm, 2022, doi:10.3390/jcm11092342_

Round 1
Reviewer 1 Report
This is a well conducted survey of small sample of Japanese patients with functional dyspepsia comparing the classification of Rome III and IV criteria. The study is well explained and the report is clearly written. The results are clearly stated with the exception of Figure 3 which is difficult to interpret.
The background should contain the actual Rome IV criteria and what is confirmed or proposed by Tack. This reference with new considerations and terminology should be incorporated in the manuscript so that language is current. The article is by Schmulson and Drossman, Journal of Neurogastroenterology and Motility 2017; 23(2): 151-163) https://doi.org/10.5056/jnm16214
The criteria have a severity score included as well now. It is not clear that it was included in the criteria. The frequency is noted, but not the severity. The description of the survey on gastrointestinal symptoms would be helpful to assure all criteria were collected since it does not appear the the Rome questionnaire was used.
Under the limitations, it should be noted that recall bias can be of concern with a survey.
Author Response
Point 1: The background should contain the actual Rome IV criteria and what is confirmed or proposed by Tack. This reference with new considerations and terminology should be incorporated in the manuscript so that language is current. The article is by Schmulson and Drossman, Journal of Neurogastroenterology and Motility 2017; 23(2): 151-163) https://doi.org/10.5056/jnm16214
Response 1: We thank the reviewer for their pertinent comments. As the reviewer mentioned, the background should contain the actual Rome IV criteria confirmed or proposed by Tack. The above reference was thus cited to present the actual Rome IV criteria and added to the list of references as Ref No. 7. We have added the sentence to Introduction (page 1, line 42-46).
Point 2: The criteria have a severity score included as well now. It is not clear that it was included in the criteria. The frequency is noted, but not the severity. The description of the survey on gastrointestinal symptoms would be helpful to assure all criteria were collected since it does not appear the the Rome questionnaire was used.
Response 2: We thank the reviewer for this very important point. As noted by the reviewer, the term “bothersome” is included in the 4 typical symptoms under the Rome IV criteria. We believe that this study should also assess severity; however, in the self-administered questionnaire, all questions related to symptoms were asked under the assumption that the symptoms were bothersome. We have added the sentence to Materials and Methods (page 3, line 114, 115).
Point 3: Under the limitations, it should be noted that recall bias can be of concern with a survey.
Response 3: Thank you for your comment. As the reviewer noted, this study obtained answers from participants using a questionnaire; therefore, we cannot dismiss the possibility that the responses were influenced by recall bias. Nonetheless, the patient-reported outcomes were analyzed; as such, we consider the results reliable. In accordance with your comment, we have added the point on recall bias will be added to the limitations (page 10, line 314-317).
Reviewer 2 Report
This study observed the change in prevalence according to the subtype of functional dyspepsia based on ROME III and ROME IV diagnostic criteria for functional dyspepsia. This simple comparative study lacks originality and seems to have considerable problems in interpreting the results.
- According to the ROME IV criteria, all epigastric symptoms after eating were classified as PDS. Therefore, the prevalence of PDS in ROME IV is increased compared to ROME III. This is a natural result due to the addition of the criteria and is not a new finding. These results cannot be different depending on race. In addition, these findings have already been confirmed in other studies cited by the authors. Furthermore, among those who fall under the overap group in ROME III, those with postprandial epigastric symptoms are classified as PDS in ROME IV. The recommendation of antiacid as the first-line treatment for these subjects is inference independent of the results of this study.
- At the beginning of the patient registration, there should be a prior investigation into the drug and medical history, such as steroids and NSAIDs, which can affect GI mucosal inflammation, GI motility drugs (prokinetics, etc) and diseases that may affect gastrointestinal motility, such as, diabetes mellitus and neurologic disease. These can cause improvement or change in symptoms of functional dyspepsia patients, so they can affect the diagnosis and classification of patients.
- There should be information on what findings from EGD were excluded as organic diseases causing epigastric symptoms. For example, chronic gastritis or duodenitis can be classified as nonorganic upper GI disease.
- How was Helicobacter pylori infection diagnosed in line 78? What is the meaning of significant GERD in line 82 and how was the diagnosis made?
- In Tables 3 and 4, the number(n) should be described.
Author Response
Point 1: According to the ROME IV criteria, all epigastric symptoms after eating were classified as PDS. Therefore, the prevalence of PDS in ROME IV is increased compared to ROME III. This is a natural result due to the addition of the criteria and is not a new finding. These results cannot be different depending on race. In addition, these findings have already been confirmed in other studies cited by the authors. Furthermore, among those who fall under the overap group in ROME III, those with postprandial epigastric symptoms are classified as PDS in ROME IV. The recommendation of antiacid as the first-line treatment for these subjects is inference independent of the results of this study.
Response 1: We thank the reviewer for the careful reading of our manuscript and for providing such useful comments. As noted by the reviewer, the incidence of postprandial distress symptoms (PDS) was higher when postprandial epigastric symptoms were diagnosed according to the Rome IV criteria of PDS by Tack et al. than when it was diagnosed according to the Rome III criteria. However, such investigation has only been reported by Tack et al.; no similar reports exist elsewhere in Japan or in other countries. Furthermore, we reported that the PDS rate increased by diagnosis according to Rome IV criteria, similar to the previous study. Nevertheless, the present study is the first to detail the clinical features of FD subtypes, and we believe that its significance lies in that combining the Rome III and IV criteria allowed accurate classification of the ambiguous “overlap” patients into PDS and EPS, thereby helping the selection of effective drug therapies.
Moreover, as noted by the reviewer, recommending the use of antiacids for overlap patients according to the Rome III criteria and to the PDS patients according to the Rome IV criteria was indeed an overstatement. We believe that a larger number of patients should be collected for additional clinical studies.
Point 2: At the beginning of the patient registration, there should be a prior investigation into the drug and medical history, such as steroids and NSAIDs, which can affect GI mucosal inflammation, GI motility drugs (prokinetics, etc) and diseases that may affect gastrointestinal motility, such as, diabetes mellitus and neurologic disease. These can cause improvement or change in symptoms of functional dyspepsia patients, so they can affect the diagnosis and classification of patients.
Response 2: We thank the reviewer for this very important point. Although this was not written in the manuscript, the present study excluded patients with organic disease by performing EGD within one year of examination before patient enrollment, and also excluded patients with clear causes to upper digestive symptoms such as malignancy and digestive ulcers, neurological diseases such as Parkinson’s disease, and metabolic diseases such as diabetes mellitus. Furthermore, patients taking drugs such as steroids, GI motility drugs, antiacids, prostaglandine preparations, antidepressants, anxiolytics and antipsychotics were also excluded. We have added this sentence to Materials and Methods (page 2, line 83-87).
Point 3: There should be information on what findings from EGD were excluded as organic diseases causing epigastric symptoms. For example, chronic gastritis or duodenitis can be classified as nonorganic upper GI disease. How was Helicobacter pylori infection diagnosed in line 78? What is the meaning of significant GERD in line 82 and how was the diagnosis made?
Response 3: We thank the reviewer for their pertinent comments. This study excluded patients with organic diseases by performing EGD within 1 year of patient enrollment. Findings such as chronic gastritis and duodenitis were classified as nonorganic upper digestive disease. Helicobacter pylori infection was diagnosed by evaluating the serum HP-IgG of patients with atrophic gastritis observed on EGD. Patients who tested positive for atrophic gastritis were excluded. Patients with clear findings of reflux gastritis on EGD or patients with predominance of GERD symptoms over dyspepsia symptoms were also excluded from the present study, and were termed “significant GERD.” In accordance with your comment, we have added this sentence to Materials and Methods (page 2, line 81, 82; page 2, line 90, 91).
Point 4: In Tables 3 and 4, the number(n) should be described.
Response 4: Thank you for your comment. As reviewer pointed out, the number (n) was added to Tables 3 and 4 accordingly.
Round 2
Reviewer 2 Report
I have no more comment.